# Two decades of change in sea star abundance at a subtidal site in Puget Sound, Washington

**Helen R. Casendino** [ID]\*, **Katherine N. McElroy, Mark H. Sorel, Thomas P. Quinn, Chelsea L. Wood**

School of Aquatic and Fishery Sciences, University of Washington, Seattle, Washington, United States of America

\* helen.casendino@gmail.com

**Data Availability Statement:** The datasets and novel code associated with this paper are available at https://github.com/wood-lab/Casendino_et_al_2023_PLoS_One. The data are also permanently archived in Dryad: Casendino, Helen et al. (2023),

## Abstract

Long-term datasets can reveal otherwise undetectable ecological trends, illuminating the historical context of contemporary ecosystem states. We used two decades (1997–2019) of scientific trawling data from a subtidal, benthic site in Puget Sound, Washington, USA to test for gradual trends and sudden shifts in total sea star abundance across 11 species. We specifically assessed whether this community responded to the sea star wasting disease (SSWD) epizootic, which began in 2013. We sampled at depths of 10, 25, 50 and 70 m near Port Madison, WA, and obtained long-term water temperature data. To account for species-level differences in SSWD susceptibility, we divided our sea star abundance data into two categories, depending on the extent to which the species is susceptible to SSWD, then conducted parallel analyses for high-susceptibility and moderate-susceptibility species. The abundance of high-susceptibility sea stars declined in 2014 across depths. In contrast, the abundance of moderate-susceptibility species trended downward throughout the years at the deepest depths– 50 and 70 m–and suddenly declined in 2006 across depths. Water temperature was positively correlated with the abundance of moderate-susceptibility species, and uncorrelated with high-susceptibility sea star abundance. The reported emergence of SSWD in Washington State in the summer of 2014 provides a plausible explanation for the subsequent decline in abundance of high-susceptibility species. However, no long-term stressors or mortality events affecting sea stars were reported in Washington State prior to these years, leaving the declines we observed in moderate-susceptibility species preceding the 2013–2015 SSWD epizootic unexplained. These results suggest that the subtidal sea star community in Port Madison is dynamic, and emphasizes the value of long-term datasets for evaluating patterns of change.

## Introduction

To detect biodiversity loss or other forms of community reorganization, species abundances must be monitored over a sufficient temporal range [1]. Long-term datasets are critical for environmental policy, management, and conservation [2, 3], providing insight by revealing shifts from one stable pattern to another [4] and revealing the effects of long-term

Two decades of change in sea star abundance at a subtidal site in Puget Sound, Washington, Dryad, Dataset, https://doi.org/10.5061/dryad.cz8w9gj7q.

**Funding:** The authors received no specific funding for this work.

environmental trends such as climate change [5]. Furthermore, long-term datasets generate baseline data that can document historical ecosystem states in the event of sudden ecological changes or catastrophes (e.g., epizootics) [6, 7]. For example, Hughes and Connell (1999) used 30 years of data from permanent reef transects to document the effects of recurrent cyclones on coral cover and recovery processes in the Great Barrier Reef [8]. Conversely, research and mitigation efforts concerning the 1989 Exxon Valdez oil spill in Prince William Sound, Alaska–responsible for the mass mortality of several marine taxa–were hampered by a dearth of baseline data [9, 10]. Similarly, when the urchin *Diadema antillarum* experienced severe die-offs in reefs throughout the Caribbean in the 1980s, inadequate historical survey data made it difficult to interpret recovery trends [11, 12].

Sea-star wasting disease (SSWD)–an epizootic that began in 2013 and affected dozens of species along ~5,000 km of North America's west coast [13, 14]–covered a larger area and resulted in greater mass mortality than other sea star wasting events in recent history [15, 16], and coincided with a marine heatwave that lasted from 2013 to 2016 [17, 18]. The sunflower star (*Pycnopodia helianthoides*) experienced substantial SSWD-associated declines [13, 14], resulting in its designation as "Critically Endangered" by the International Union for Conservation of Nature in 2020 [19]. Miner et al. (2018) observed precipitous declines in ochre star (*Pisaster ochraceus*) abundance from southern California to southeastern Alaska in 2014 and 2015 [20]; in Oregon, Menge et al. (2016) found that *P. ochraceus* biomass declined 80–99% during the same period [15], though *P. ochraceus* recruitment has increased in the years following SSWD's onset [21].

SSWD altered intertidal and subtidal community structure across the west coast of North America. *Pycnopodia helianthoides* and *Pisaster ochraceus* are keystone predators that exert strong top-down control of other species in subtidal and intertidal habitats [22–26]. Schultz et al. (2016) found that the SSWD-induced decline of predatory *P. helianthoides* was followed by a four-fold increase in green sea urchin (*Strongylocentrotus droebachiensis*) abundance in the Salish Sea, and a decline in kelp cover from 4% (± 10%) of the study area to < 1% (± 2%) [24]. Changes to kelp forests in California during this time period also indicated the influence of *P. helianthoides* declines on urchin-mediated regime shifts [17, 27]. Shifts in ecosystem dynamics attributable to SSWD also include changes to sea star community structure. Monte-cino-Latorre et al. (2016) monitored sea star density in depths of 6–18 m [28] in scuba diving surveys and observed that declines in pink sea stars (*Pisaster brevispinus*) and *Pycnopodia helianthoides* were coupled with an increase in leather stars (*Dermasterias imbricata*) but no change in vermillion stars (*Mediaster* spp.) following the 2013 outbreak. Additionally, Kay et al. (2019) observed that mottled stars (*Evasterias troschelii*) became dominant following SSWD's onset as formerly dominant *Pisaster ochraceus* declined, potentially due to competitive release [29].

The etiological agent responsible for SSWD is currently unknown; a pathogen previously linked to SSWD [30] was later found in sea stars with no signs of disease [13, 31]. Studies conducted in situ and in laboratories have linked increased temperatures to wasting prevalence [32], sea star mortality [14], and pace of infection progression [33, 34]. However, other studies have found temperature to be uncorrelated [35] or negatively correlated [15, 30] with SSWD. Thus, it is unlikely that temperature is a proximate cause of wasting in sea stars, though it may play an indirect or interactive role with a causal agent [13, 36]. Lab experiments by Aquino et al. (2021) suggested that body lesions characteristic of SSWD could be a physiological response to low oxygen conditions created by organic matter-fueled microbial activity on the sea star's outer surface, supported by additional in situ observations correlating chlorophyll-*a* and sea star mortality between 2014 and 2019 [36]. However, it is difficult to imagine hypoxia being a cause for SSWD in the intertidal zone, where wave action creates high-oxygen

conditions [37]. Whatever the etiologic agent, SSWD susceptibility appears to vary among sea star species [21, 38]. Schiebelhut et al. (2022) aggregated findings on the level of SSWD-associated mortality experienced by several sea star species [21]. Based on frequency of wasting observations or SSWD-associated population declines, the following species were found to have faced high SSWD-associated mortality: *Pisaster brevispinus*, *E. troschelii*, *Pycnopodia helianthoides*, *Solaster stimpsoni*, *S. dawsoni*, and others. Other species exhibited lower SSWD-associated mortality: *Crossaster papposus*, *D. imbricata*, *Hippasteria spinosa*, *Mediaster aequalis*, *Luidia foliolata*, and others.

Most studies on the impacts of SSWD concern intertidal habitats, but some examined impacts in subtidal habitats. Harvell et al. (2019) observed significant declines in both nearshore abundance (3–15 m) and offshore biomass (55–1280 m) of *Pycnopodia helianthoides* after 2013 in Washington, Oregon, and California [14], while Montecino-Latorre et al. (2016) found that *Pisaster brevispinus* and *Pycnopodia helianthoides* in shallow subtidal depths (6–18 m) declined following the 2013 outbreak [28]–findings consistent with patterns observed in the intertidal [13]. Aside from these examples, observations of SSWD impacts in subtidal habitats are rare.

Our objective was to evaluate whether the total abundance of subtidal sea stars has changed over the past two decades at Port Madison, Washington, USA, a long-term study site. We predicted declines in sea star density coincident with the 2013–2015 SSWD event, with a possible recovery from 2016 onwards [35]. We further predicted that these declines would be steeper for species with high reported susceptibility to SSWD relative to species with moderate reported susceptibility, and that warm temperatures would be associated with declines, as might result from increased microbial activity or eroded sea star immune defenses [14, 36, 39]. We combined a 20-year trawling dataset from Port Madison with water temperature data from Puget Sound to test our hypotheses regarding subtidal sea star community trends. Our study's long temporal scale gives rare insight into the response of these subtidal populations not only to the SSWD epizootic, but to broader environmental changes of the past two decades.

## Materials and methods

### Sea star sampling

In each year from 1997 to 2019 (except for 1998), we chartered a vessel equipped with a Southern California Coastal Waters Research Program (SCCWRP) otter trawl net for an annual research cruise near Port Madison, a bay on the west side of Puget Sound, WA [40]. This sampling location experiences a moderate amount of human impact; it is close to a major city (Fig 1) and an outfall of the Suquamish Wastewater Treatment Plant [41]. However, commercial bottom trawling has not been permitted during the period of our sampling [42] and the nearby shoreline is only moderately developed for residential properties [40]. The cruise was designed as a teaching experience for students enrolled in a University of Washington course (Fisheries Ecology / FISH 312) [43]. The sampling protocol for these field trips received a full review every three years from the School of Aquatic and Fishery Sciences and the Institutional Animal Care and Use Committee of the University of Washington (IACUC protocol # 2442–13). In addition, the field trips were annually reviewed and permitted by the Washington Department of Fish and Wildlife (SCP 22–102). The vessel, which ran each year for two days in mid-May (between 10 and 18 May), towed the SCCWRP net along the bottom in set trawling locations corresponding to four depths: 10, 25, 50 and 70 m (Fig 1). Each year, five trawls took place at each depth, corresponding to discrete time periods: early morning (~6:00–8:00), morning (~10:00–12:00), afternoon (~15:00–17:00), evening (~20:00–22:00), and night (~1:00–3:00).

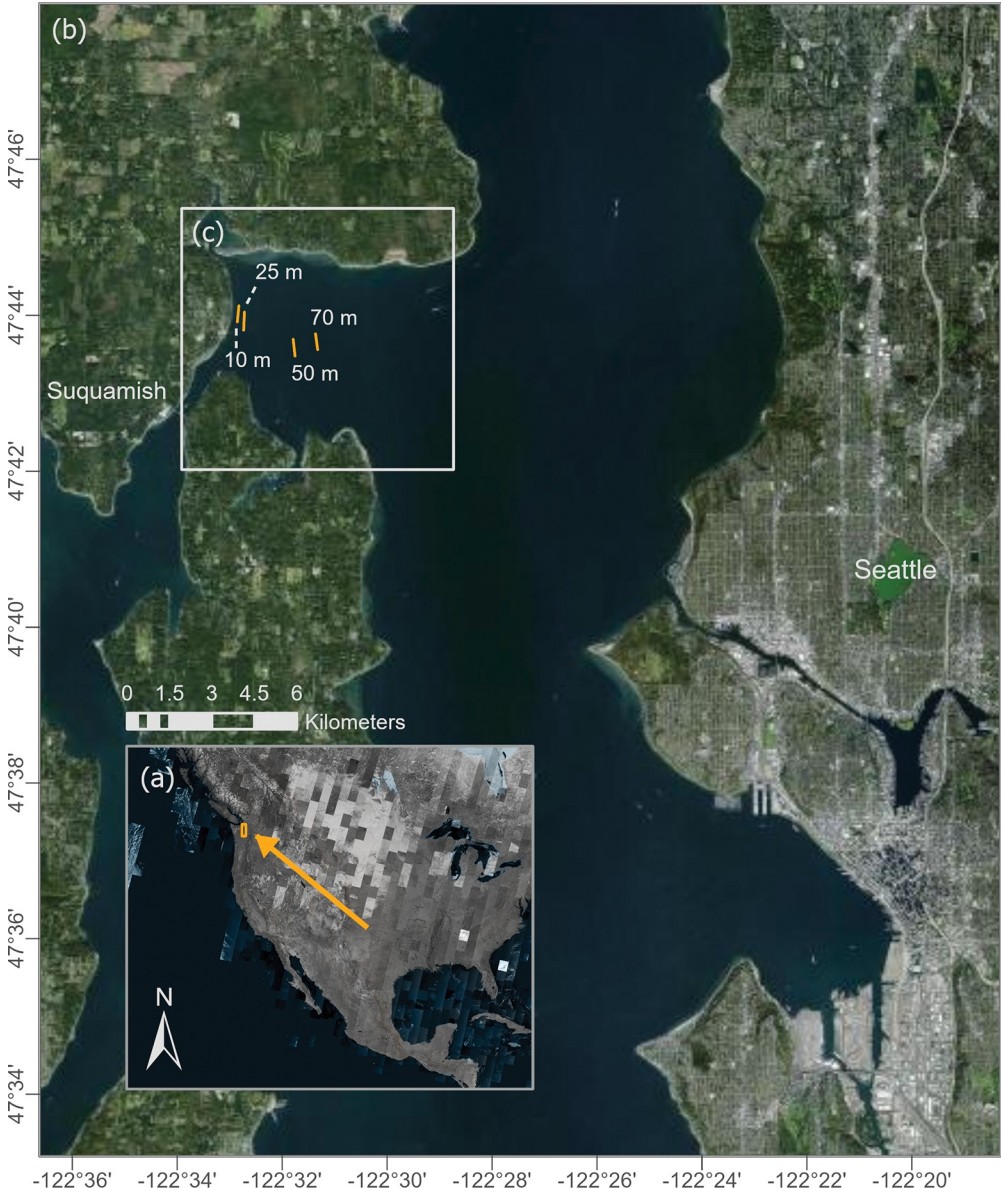

**Fig 1.** Study site at different scales: North America (a), Puget Sound (b), and Port Madison (c). Map of study site and trawling locations. The orange rectangle in the Inset (a) shows the map's boundaries. The orange lines at our sampling site in Port Madison (c) represent the standard distances and locations of sampling trawls at each depth.

Each trawl sampled benthic habitat for ~5 min over 370 m (mean = 372, range = 367 to 538 m). The net's opening was 3.5 m wide and 1.0 m high, with 35 mm mesh. Following each trawl, all sea stars caught were identified to species, counted, recorded, and released. The sampling sites were selected for the ease of maintaining a constant depth over the trawl distance, and the absence of rocks or other obstructions rather than any other specific attributes. The bottom in this area is primarily fine sand and silt, as is typical in Puget Sound at these depths [44].

We categorized our data by the relative susceptibility of sea star species to SSWD based on Schiebelhut et al. (2022), who classified the level of SSWD-associated mortality faced by several sea star species [21]. Of the species present in our data, five were categorized by Schiebelhut et al. as "high mortality" (*Solaster stimpsoni*, *S. dawsoni*, *Pycnopodia helianthoides*, *Pisaster*

*brevispinus*, *E. troschelii*), which we categorized as "high-susceptibility" species. Two additional species were categorized by Schiebelhut et al. as "noticeable mortality" (*D. imbricata*, *Henricia leviuscula*) and four as "likely affected" (*Mediaster aequalis*, *Luidia foliolata*, *Crossaster papposus*, *Hippasteria spinosa*), all of which we categorized as "moderate-susceptibility" species, which we assumed would experience lower SSWD-associated mortality than high-susceptibility species. Subsequent processing and analysis of abundance data was conducted in parallel for these two data groupings. We did not analyze species-specific trends in our data due to insufficient replication at the level of individual species.

To estimate sea star density, we calculated the area swept by multiplying the net width (3.5 m) by each trawl's distance (measured by GPS). We then divided the total number of sea stars (within each susceptibility category) found in each trawl tow (corresponding to a single depth and time) by the area swept (and multiplied by 1000 for readability) to calculate total sea star density (specimens per 1000 m$^2$) of high-susceptibility and moderate-susceptibility species.

## Water temperature

We obtained data on water temperature collected from 1999–2017 by the Washington State Department of Ecology at a site ~9 km southeast of our trawling sites (47.66001˚, –122.4417˚). A thermistor measured temperature throughout the water column, to a median maximum depth of 43.5 m. We calculated the average of median temperatures from sampling dates in March, April, and May of each year in the dataset (thermistor maximum depth between 20 and 60 m) to estimate annual temperature averages.

## Analysis of directional change over time

We used generalized linear mixed models to explore trends in the catch across years and depths, as well as relationships with water temperature. Filtered to include high-susceptibility or moderate-susceptibility species only, the total catch of sea stars, $c_{i,d,t}$ in a given trawl conducted at depth $d$ at time of day $i$ in year $t$ was assumed to be negative binomially distributed around expected catch $\mu_{d,t}$ with dispersion parameter $k$. The expected catch $\mu_{d,t}$ was modeled on the log scale as a function of year- and depth-specific design covariates $x_{d,t}$ and coefficient vector $\beta$, an offset for trawl area $a$, and random effects for year and depth $\varepsilon_{d,t}$, which follow a stationary first-order autoregressive process across years at each depth:

$$log\,(\mu_{d,y}) = x_{d,t}\,\beta + log\,(a_{i,d,t}) + \varepsilon_{d,t};\ \varepsilon_{d,t} \sim \Sigma_{ar1}$$

$$c_{i,d,t} \sim Negative\ Binomial(\mu_{d,t}, k)$$

where $\Sigma_{ar1}$ is the standard autoregressive covariance matrix. A common innovation variance and autocorrelation coefficient in the autoregressive covariance matrix were used across depths. We treated trawls conducted at different depths and times of day as replicates (~20 replicates per year). While trawls were conducted in roughly the same location for each depth sampled, natural variability in each trawl's path made overlap unlikely. We confirmed this using a generalized linear model where expected catch was modeled as a function of trawl number (corresponding to time of day) at each depth. We observed no significant relationship between trawl number and expected total sea star catch (estimate = 0.03, ± SE = 0.12, $p$ = 0.79).

We compared data support for a null model with no effect of year and eight models with different specifications of fixed effects of year to evaluate evidence of trends and step changes in the abundance of high-susceptibility and moderate-susceptibility species (Table 1). In all models, a unique intercept was fit for each depth (10, 25, 50 and 70 m). To evaluate support for a directional shift in sea star catch coinciding with the detection of the SSWD outbreak in

**Table 1. Support for multiple models of sea star catch in bottom trawls in Puget Sound.**

| Model [high-susceptibility] | Df | AICc | Δ AICc | Model [moderate-susceptibility] | Df | AICc | Δ AICc |
|---|---|---|---|---|---|---|---|
| Depth + PrePost | 8 | 496.00 | 0.00 | Depth + Year × Depth | 11 | 402.46 | 0.00 |
| Depth + PrePost × Year | 9 | 496.09 | 0.09 | Depth + PrePost × Year × Depth | 15 | 406.78 | 4.32 |
| Depth + PrePost + Year | 9 | 497.01 | 1.01 | Depth + PrePost × Depth + Year × Depth | 15 | 407.22 | 4.76 |
| Depth + Year | 8 | 499.51 | 3.51 | Depth + PrePost × Depth | 11 | 416.28 | 13.82 |
| Depth + PrePost × Depth | 11 | 499.74 | 3.74 | Depth + PrePost × Year | 9 | 416.59 | 14.13 |
| Depth + PrePost × Year × Depth | 15 | 503.66 | 7.66 | Depth + PrePost + Year | 9 | 417.29 | 14.83 |
| Depth + Year × Depth | 11 | 503.94 | 7.94 | Depth + Year | 8 | 417.42 | 14.96 |
| Depth + PrePost × Depth + Year × Depth | 15 | 504.50 | 8.50 | Depth | 7 | 418.13 | 15.67 |
| Depth | 7 | 508.84 | 12.84 | Depth + PrePost | 8 | 420.16 | 17.70 |

Models of high-susceptibility sea star catch are represented by the table's left four columns, while models of moderate-susceptibility sea star catch are represented by the right four columns. Depth is a categorical variable (10, 25, 50 and 75 m), PrePost is a categorical variable indicating whether the year was before or after the SSWD outbreak in 2013, and Year is a continuous covariate for year. Df is the degrees of freedom of the model (equal to the number of parameters +2), AICc is Akaike's Information Criterion corrected for small sample sizes, and ΔAICc is the difference from the lowest AICc. In the model specifications, × indicates an interaction (e.g. PrePost × Depth is a unique step change in each depth).

2014, we fit models where year was discretized into a factor to indicate the period pre-SSWD (1997–2013) and post-SSWD (2014–2019). To evaluate support for a continuous trend in abundance, we fit models where year was a continuous covariate. Other models allowed for different trends (i.e., slopes) before and after 2014 (post-SSWD). We also fit models with a step change in 2014 as well as a continuous trend over the course of the study. For each model hypothesis concerning the effect of year on sea star abundance, we fit a model where year effects were common across depths, and another model where year effects were unique across depths. We compared data support for models based on AICc (AIC corrected for small sample sizes using the *AICc()* function of the "MuMIn" package) [45, 46], to explore evidence of trends throughout the study period and step changes in 2014.

Using the best-supported specification of fixed and random effects for each susceptibility group, we assessed the effect of water temperature on catch of high-susceptibility and moderate-susceptibility species. We explored this effect in separate models from the primary models of year effects, because temperature measurements were not taken for three sampling years (1997, 2018, 2019). All models were fit using the glmmTMB package [47]. We assessed goodness of fit using the following residual diagnostic tests in the DHARMa package: KS test, dispersion test, outlier test, and residuals vs. predicted [48].

## Changepoint analysis

To look for evidence of a shift in abundance that preceded the first detection of SSWD in 2013 or that lagged years behind that first detection, we filtered our data to include the high-susceptibility or moderate-susceptibility species only, and used a vector of average annual sea star densities across depth to evaluate the presence of changepoint(s) in sea star density with the *processStream()* function of the "cpm" package [49, 50]. The *processStream()* function operates using sequential changepoint detection [49]. We chose the *processStream()* function (a Phase II method) because it accounts for the possibility of a high number of changepoints [51]. The CPM type used for the *processStream()* function was the Mann-Whitney test statistic, a nonparametric approach to detect location shifts in non-gaussian sequences [49]. Using these methods, we identified the changepoints in average sea star density both across all depths and then within each depth category.

**Table 2. Mean abundance per trawl of sea star species.**

| Species | 10 m | 25 m | 50 m | 70 m | All depths |
|---|---|---|---|---|---|
| *Mediaster aequalis* | 0.009 | 0.019 | 0.046 | 0.630 | 0.177 |
| *Solaster stimpsoni* | 0.084 | 0.028 | 0.157 | 0.306 | 0.144 |
| *Pycnopodia helianthoides* | 0.187 | 0.121 | 0.046 | 0.000 | 0.088 |
| *Luidia foliolata* | 0.000 | 0.000 | 0.074 | 0.093 | 0.042 |
| *Crossaster papposus* | 0.019 | 0.009 | 0.000 | 0.037 | 0.016 |
| *Hippasteria spinosa* | 0.000 | 0.000 | 0.009 | 0.037 | 0.012 |
| *Dermasterias imbricata* | 0.009 | 0.028 | 0.000 | 0.000 | 0.009 |
| *Pisaster brevispinus* | 0.019 | 0.009 | 0.009 | 0.000 | 0.009 |
| *Solaster dawsoni* | 0.009 | 0.009 | 0.00 | 0.019 | 0.009 |
| *Evasterias troschelii* | 0.019 | 0.000 | 0.000 | 0.000 | 0.005 |
| *Henricia leviuscula* | 0.000 | 0.009 | 0.000 | 0.000 | 0.002 |

Average sea star abundance per trawl by species, listed in order of overall abundance, based on 5 tows over a 24-h period each year at each depth from 1997–2019.

## Results

Over the 22 years sampled, 430 trawl tows were completed, and 269 sea stars from 11 species were caught across all depths and years: 38 individuals at 10 m, 37 at 25 m, 37 at 50 m, and 157 at 70 m (mean = 0.00048 sea stars per $m^2$ of bottom trawled). This density was similar to that estimated by bottom trawls (55–1280 m) conducted by Harvell et al. (2019) [14] measuring *P. helianthoides* biomass on Washington's outer coast (0–0.00025 sea stars per $m^2$ of bottom trawled; converted from units of kg per ha using an estimated average mass of *P. helianthoides* from Washington's outer coast of 2 kg based on Juorio and Robertson 1977) [52]. Many of the sea star species affected by SSWD in the Salish Sea were represented in our dataset (Table 2; Fig 2) [28], including *Pycnopodia helianthoides*, *Pisaster ochraceus*, and *S. stimpsoni*.

The best supported linear model of high-susceptibility sea star catch included a step change in 2014 that was common across depths (Table 1). In 1997, the first year of the study, the expected sea star density (annual catch per 1000 $m^2$ of bottom trawled) predicted by our model was slightly higher at shallower depths (10 m: estimate = 0.20, 95% CI = 0.079–0.52) compared to deeper depths (25 m: estimate = 0.12, 95% CI = 0.042–0.34; 50 m: estimate = 0.077, 95% CI = 0.025–0.24; 70 m: estimate = 0.10, 95% CI = 0.034–0.32). There was evidence of a decrease in sea star catch after 2014 across depths (estimate = -2.77, ± SE = 0.81, p < 0.001; Fig 3). The average spring temperature median was 9.4° C (range: 7.4° to 13.6° C). Temperature was not associated with catch (estimate = -0.034, ± SE = 0.40, *p* = 0.93). No evidence of lack of fit was detected in any of the model fits based on residual diagnostic tests.

The linear model that best explained total catch of moderate-susceptibility species included one unique continuous trend across years at each depth and no step change in 2014 (Table 1). In 1997, the expected sea star density predicted by our model was highest at 70 m (estimate = 8.04, 95% CI = 2.37–27.3) and was considerably lower at 10 m (estimate = 0.0064, 95% CI = 0.00031–0.13), 25 m (estimate = 0.0093, 95% CI = 0.00074–0.12) and 50 m (estimate = 0.43, 95% CI = 0.081–2.29). There was no evidence of a decreasing trend in catch at 10 m (estimate = 0.085, ± SE = 0.098, p = 0.39) or 25 m (estimate = 0.096, ± SE = 0.082, p = 0.24). However, there was evidence of decreasing trends in catch at 50 m (estimate = -0.24, ± SE = 0.10, p = 0.024) and 70 m (estimate = -0.34, ± SE = 0.074, p < 0.001) (Fig 4). Contrary to our prediction, temperature was positively associated with catch (estimate = 1.20, ± SE = 0.33, p < 0.001). No evidence of lack of fit was detected in any of the model fits based on residual diagnostic tests.

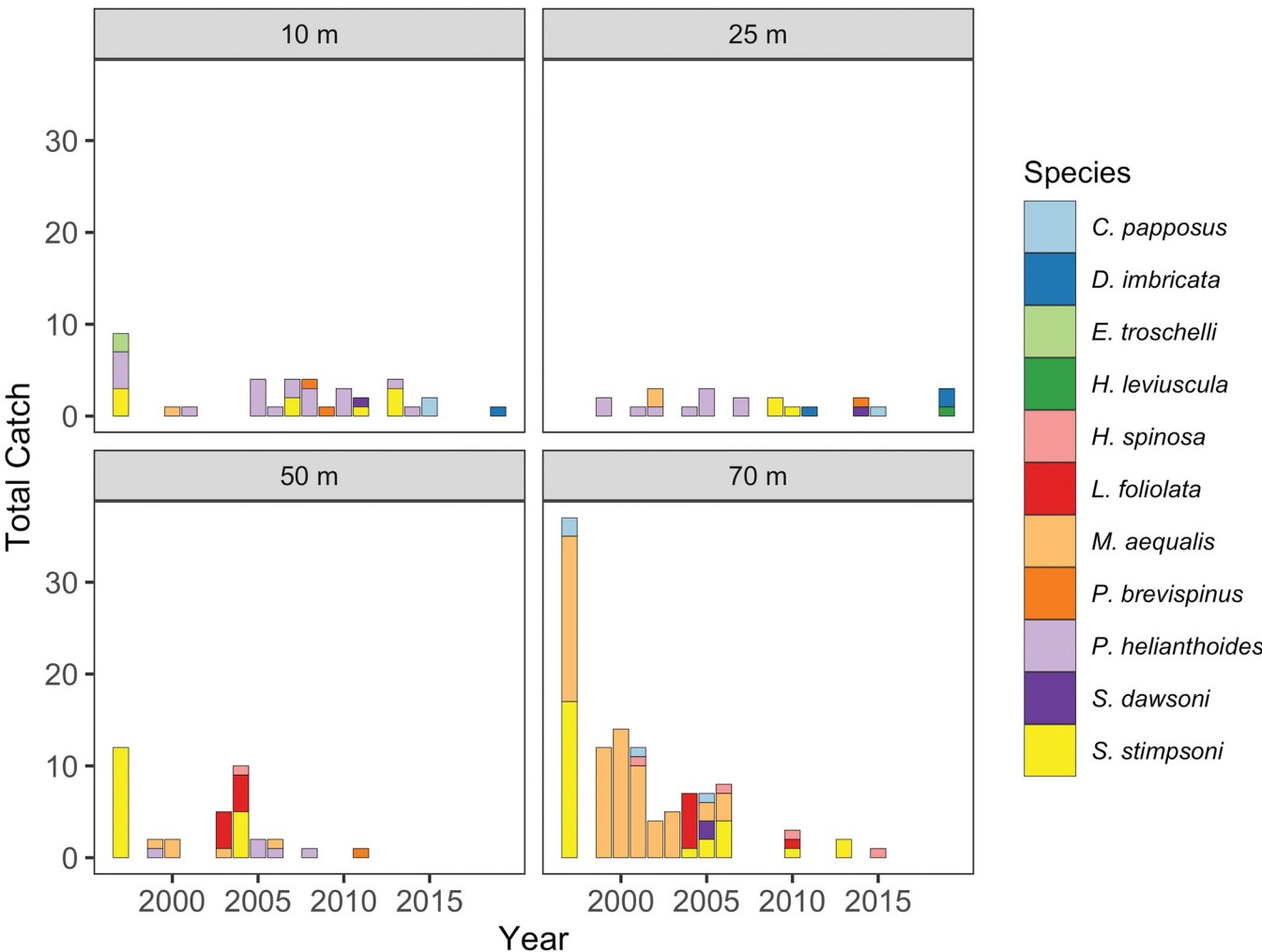

**Fig 2. Observed sea star catch by species at four depths from 1997 through 2019.** Sea star catch per year, summarized per species over 5 tows within a 24-h period each year at each depth from 1997–2019.

Within and across all depth categories, we observed no significant changepoints in high-susceptibility sea star catch (Fig 5). We observed a significant changepoint in moderate-susceptibility sea star catch in 2006 when we evaluated our data across all depths (Fig 6). When depths were analyzed individually for this subset of species, a changepoint was observed in 2006 at 70 m. Following each changepoint of moderate-susceptibility sea star catch, sea star abundance declined (Fig 6).

## Discussion

Consistent with our predictions and previous reports in Washington State [14, 32, 53], our analyses suggest that the density of high-susceptibility sea stars was lower after 2013 (Fig 3). Unexpectedly, our analyses also suggest that moderate-susceptibility sea stars experienced long-term declines throughout the study period in deeper water (50 and 70 m) (Fig 4), and a sudden decline in 2006 at 70 m and across depths (Fig 6). Temperature was positively associated with moderate-susceptibility sea star abundance.

Our results suggest that, of the sea star species included in our sample, those observed in other studies to be most impacted by SSWD decreased in abundance following the onset of

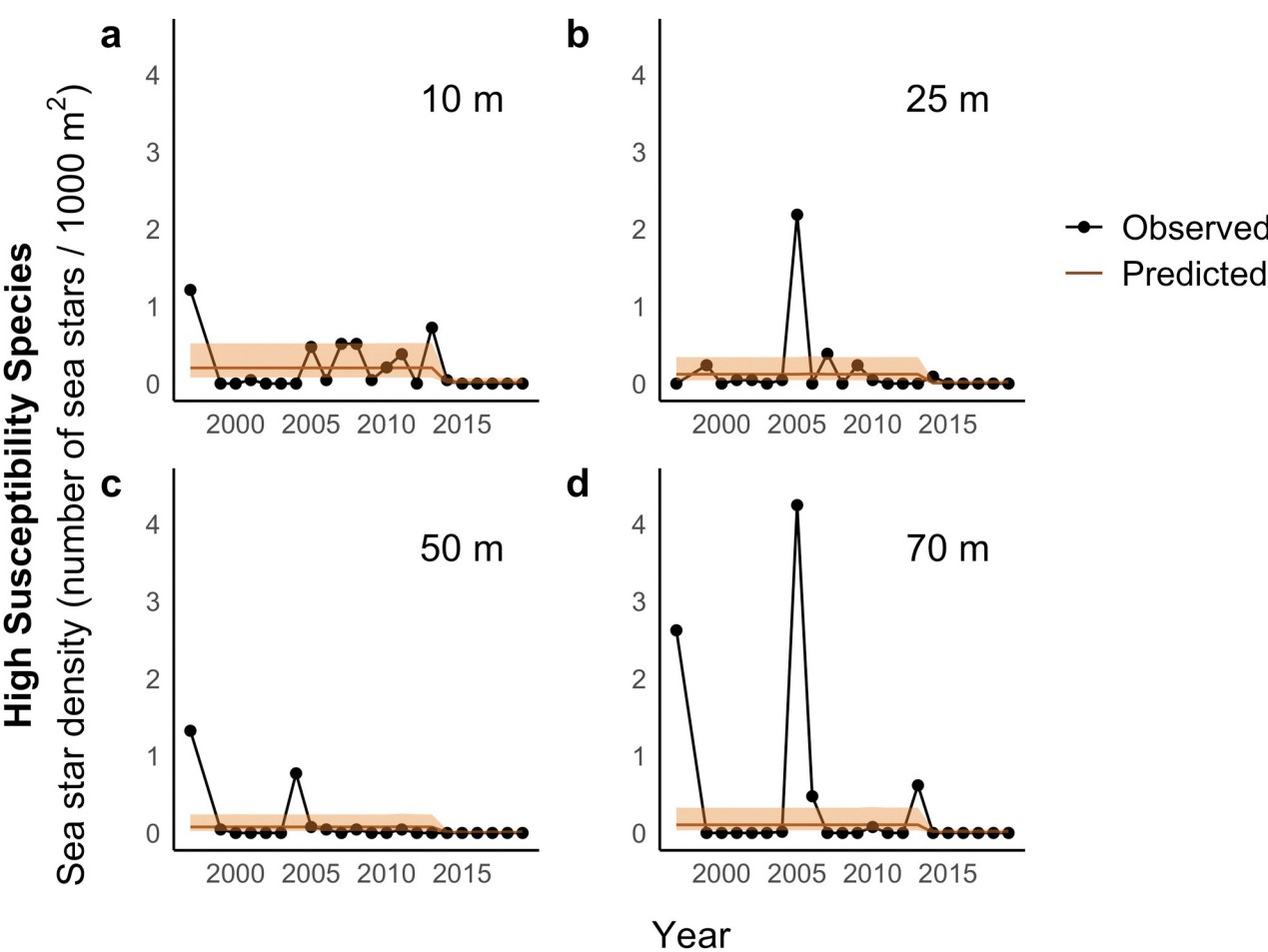

**Fig 3. Observed and predicted high-susceptibility sea star density (number of sea stars / 1000 m²) at four depths from 1997 through 2019.** Black points represent the annual catch of high-susceptibility sea stars per 1000 m² of bottom trawled, averaged over five samples conducted at different times of day. Orange lines represent expected sea star density based on a generalized linear mixed effects model with a step change between the periods 1997–2013 and 2014–2019 that is consistent across depth, and autocorrelated random effects of year. The predictions shown are made with random effects of year set to zero to highlight the trends over years. The orange shaded areas represent 95% confidence intervals around the expected annual mean density.

SSWD in 2013–14. These findings match other accounts of declines in the region; on Washington's Pacific coast, Harvell et al. (2019) observed a 99.2% biomass decrease of *Pycnopodia helianthoides* between 2013 and 2015 [14]. Nearer to this study's sampling site, in south Puget Sound, two separate SSWD outbreaks occurred in the winter and summer of 2014 [32]. It is likely that *P. helianthoides* and *S. stimpsoni* contributed most to the sudden decline we observed post-SSWD, given their relatively high abundance in our samples (Table 2; Fig 2). While previous findings link heightened SSWD susceptibility to shallow depths [21, 54], our results align with reports of SSWD-associated mortality across intertidal and subtidal habitats [14, 28, 32, 55].

Our findings also indicated long-term declines in the abundance of moderate-susceptibility species throughout the sampling period at 50 and 70 m (Fig 4). Notably, the best supported linear model of moderate-susceptibility sea star catch did not include a step change in 2014, as these species were less impacted by SSWD's onset. We also observed a changepoint at 70 m and across all depths, suggesting that moderate-susceptibility sea star density suddenly declined in 2006, seven years before the documented SSWD epizootic (Fig 6). This sharp

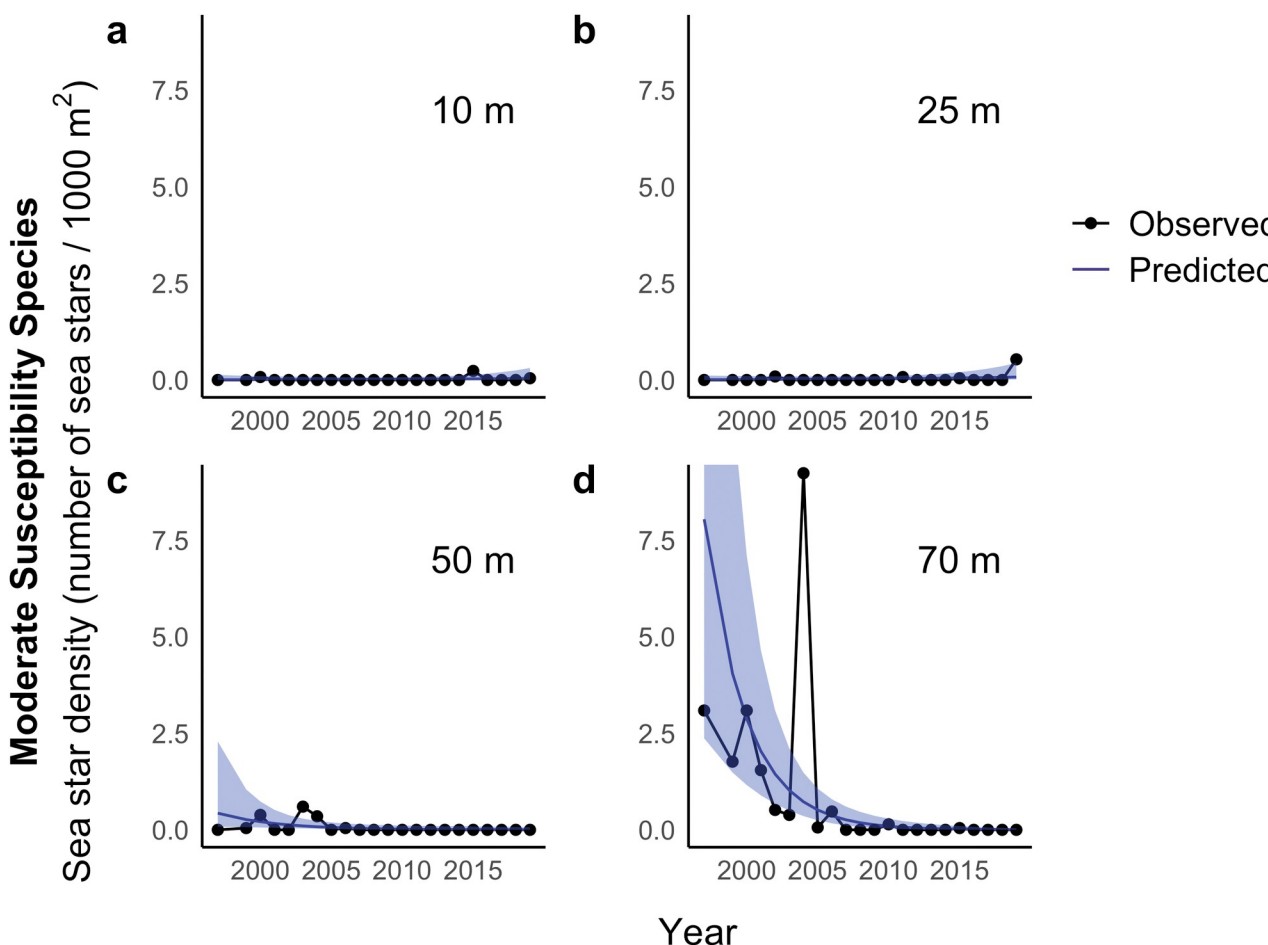

**Fig 4. Observed and predicted moderate-susceptibility sea star density (number of sea stars / 1000 m²) at four depths from 1997 through 2019.** Black points represent the annual catch of moderate-susceptibility sea stars per 1000 m² of bottom trawled, averaged over five samples conducted at different times of day. Blue lines represent expected sea star density based on a generalized linear mixed effects model with a unique intercept and slope for each depth and autocorrelated random effects of year. The predictions shown are made with random effects of year set to zero to highlight the trends over years. The blue shaded areas represent 95% confidence intervals around the expected annual mean density (95% CI upper limit for 70 m: 27.3).

decline may be attributable to a localized disease outbreak in Port Madison, as was experienced by *Pisaster ochraceus* in Barkley Sound (British Columbia, Canada) during the summer of 2008, when Bates et al. (2009) observed an increase in *P. ochraceus* bearing signs of wasting similar to those observed during the 2013 SSWD epizootic [34]. Regional outbreaks of sea star wasting–operating on smaller geographic scales and affecting fewer species than the SSWD epizootic [16, 31, 38]–have been documented for decades [54, 56–59], and one such event may have occurred in Port Madison in 2006.

There is little evidence of persistent disease in sea stars before the SSWD epizootic that would explain the protracted declines we observed at 50 and 70 m in moderate-susceptibility species, though McPherson et al. (2021) found that *Pycnopodia helianthoides* density decreased continuously from 2003–2018 but did not identify a cause for the decline [17]. Aquino et al. (2021) and Hewson (2021) reported that chlorophyll-*a*, stratification [36, 60], and upwelling may be linked to signs of wasting in sea stars, noting that these variables influence the amount of dissolved organic matter and subsequent microbial activity at a sea star's surface. Decreasing frequency of phytoplankton blooms and average chlorophyll-*a* in Puget Sound since 1999 [61,

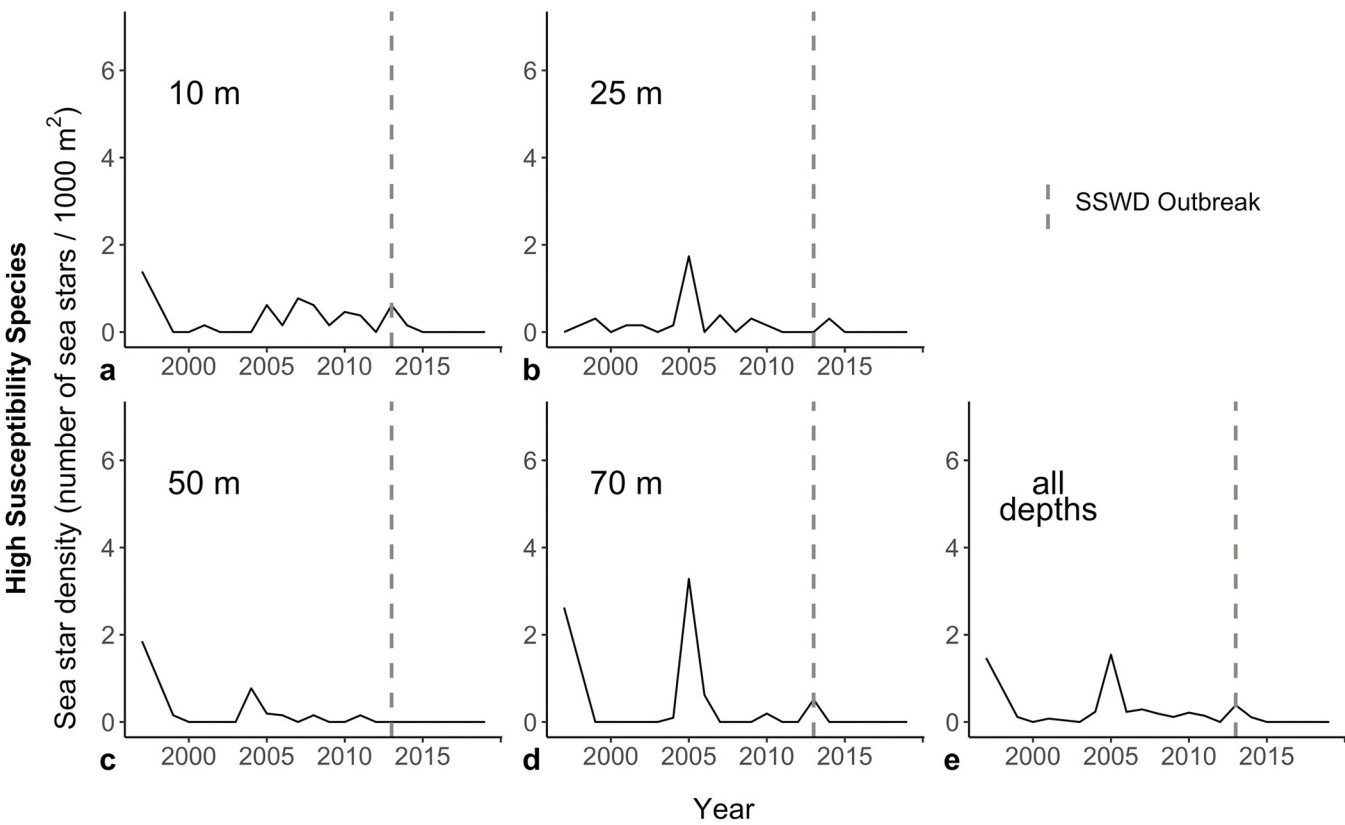

**Fig 5. Change in mean high-susceptibility sea star density (number of sea stars / 1000 m² ) from 1997 to 2019 at different depths.** The dashed gray line demarcates 2013, the first year of the sea star wasting disease epizootic in the Salish Sea.

62] makes it unlikely that these variables caused wasting (and subsequent declines) in sea stars in Port Madison. However, a decrease in overall productivity could affect sea stars through trophic links, reducing resource availability [61, 62]. It is possible that the observed long-term declines in sea star abundance resulted from our repeated bottom trawling at a study site over multiple years but this is unlikely. Natural variability in each trawl's path would make it unlikely for the same area to be trawled twice, and we found no evidence of serial depletion across consecutive trawls within a given sampling year.

We expected high temperatures to be correlated with low sea star density across susceptibility groups, either because of high microbial activity and subsequent oxygen depletion [36] or higher host metabolic demands, stress, and lower immunity [34, 63, 64]. Instead, we found a positive correlation between abundance of moderate-susceptibility sea star species and temperature, further indicating the inconsistent relationship between sea star mortality and temperature [15, 33–35]. It should also be noted that all sea stars sampled were possibly exposed to SSWD-associated pathogens. While outside of the scope of this study, complex interactions of environmental variables and their effects on wasting prevalence in sea stars—isolated from the influence of any pathogens linked to SSWD—warrant further investigation. For species less susceptible to SSWD, the long-term effects of environmental and/or anthropogenic factors seemed to have overwhelmed the effects of the 2013 epizootic.

The decline of sea star abundance in subtidal zones has important implications for the organization of those communities. Subtidal sea stars substantially influence the abundance and size distribution of numerous prey species [65–67]. Though Gaymer et al. (2004) only looked

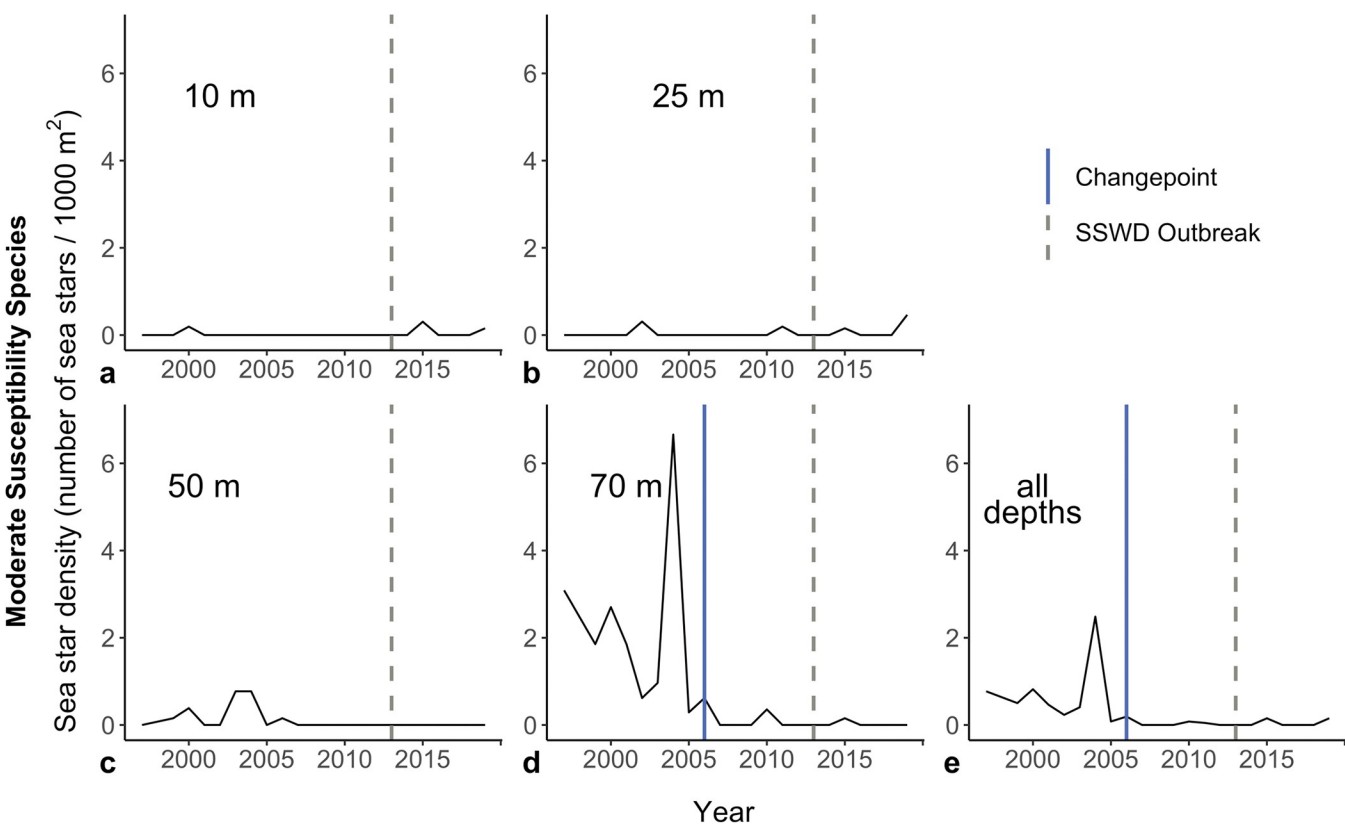

**Fig 6. Change in mean moderate-susceptibility sea star density (number of sea stars / 1000 m²) from 1997 to 2019 at different depths.** The dashed gray line demarcates 2013, the first year of the sea star wasting disease epizootic in the Salish Sea. The solid blue line demarcates changepoints in mean density of moderate-susceptibility sea stars at depth.

at depths of 8–11 m, their findings demonstrate the myriad functions that sea stars serve in their community, including predatory control of blue mussels (*Mytilus edulis*) among other marine taxa [66]. Changes in the subtidal zone also have implications for intertidal communities. High connectivity between intertidal and subtidal species occurs in estuaries [68] and salt marshes [69]. In marine ecosystems, subtidal and intertidal food webs may be linked via microalgal carbon [70]. Catastrophes that eliminate sea stars from deeper subtidal depths threaten the stable community structure sea stars provide, risking the loss of subtidal and intertidal ecosystem services.

Understanding the implications of SSWD and other changes in subtidal sea star populations is vital, considering their great ecological importance in top-down regulation of subtidal environments [17, 27, 35]. The perspective given by long-term data offers a more reliable picture of how communities have responded, and may continue to respond, to major ecological events. This work suggests that some subtidal sea star species experienced gradual and steep declines not strictly attributable to the SSWD epizootic of 2013; whether this decline was part of a long-term population cycle or a response to environmental change warrants further investigation.

## Acknowledgments

The sampling described herein was supported as part of the teaching program at the University of Washington's School of Aquatic and Fishery Sciences (SAFS), and we are grateful for SAFS'

commitment to experiential learning. The vessel from which almost all sampling took place was owned and operated by Charles Eaton, and we appreciate his skillful operation and assistance with species identification, as well as the help from the dozens of teaching assistants and hundreds of students over the years. We also thank the crew of the R/V Rachel Carson, which is the current sampling platform and contributed 2019 data. Mark Scheuerell assisted with the changepoint analysis and Monica Moritsch provided perspective on sea star wasting disease. Chelsea L. Wood was supported by a Sloan Research Fellowship from the Alfred P. Sloan Foundation, a grant from the National Science Foundation (OCE-1829509), and a University of Washington Innovation Award.

## Author Contributions

**Conceptualization:** Helen R. Casendino, Katherine N. McElroy, Chelsea L. Wood.

**Data curation:** Thomas P. Quinn.

**Formal analysis:** Helen R. Casendino, Mark H. Sorel, Chelsea L. Wood.

**Investigation:** Thomas P. Quinn.

**Visualization:** Helen R. Casendino, Mark H. Sorel.

**Writing – original draft:** Helen R. Casendino.

**Writing – review & editing:** Helen R. Casendino, Katherine N. McElroy, Mark H. Sorel, Thomas P. Quinn, Chelsea L. Wood.

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
