## [Decision Letter · Decision Letter 0]

14 Feb 2023

PONE-D-22-25271

Two decades of change in sea star abundance at a subtidal site in Puget Sound, Washington

PLOS ONE

Dear Dr. Casendino,

Thank you for submitting your manuscript to PLOS ONE. After careful consideration, we feel that it has merit but does not fully meet PLOS ONE’s publication criteria as it currently stands. Therefore, we invite you to submit a revised version of the manuscript that addresses the points raised during the review process.

The two reviewers have made many good suggestions for improving your manuscript. I'll point out that both of them were not satisfied with lumping everything into total sea stars, and wanted species specific trends wherever possible. Given the low densities (Table 2), the absolute numbers must be low also. A figure with the species-specific numbers could be a good idea. The Github link to the data that you provided isn't complete, so I couldn't look at the direct numbers. If I am doing the math correctly converting from Table 2, lots of those values are 1, 2, etc. I wonder if you could do some kind of heat map with the absolute values? That might be an easier way to display the species-specific trends in an informative manner. 

We look forward to receiving your revised manuscript.

Kind regards,

Erik V. Thuesen, Ph.D.

Academic Editor

PLOS ONE

4. We note that [Figure 1] in your submission contain [map/satellite] images which may be copyrighted. All PLOS content is published under the Creative Commons Attribution License (CC BY 4.0), which means that the manuscript, images, and Supporting Information files will be freely available online, and any third party is permitted to access, download, copy, distribute, and use these materials in any way, even commercially, with proper attribution. For these reasons, we cannot publish previously copyrighted maps or satellite images created using proprietary data, such as Google software (Google Maps, Street View, and Earth). For more information, see our copyright guidelines: http://journals.plos.org/plosone/s/licenses-and-copyright.

a. You may seek permission from the original copyright holder of Figure(s) [#] to publish the content specifically under the CC BY 4.0 license. 

Natural Earth (public domain): http://www.naturalearthdata.com/.

Reviewers' comments:

Reviewer's Responses to Questions

**Comments to the Author**

1. Is the manuscript technically sound, and do the data support the conclusions?

Reviewer #1: Partly

Reviewer #2: Partly

2. Has the statistical analysis been performed appropriately and rigorously? 

Reviewer #1: Yes

Reviewer #2: Yes

3. Have the authors made all data underlying the findings in their manuscript fully available?

Reviewer #1: Yes

Reviewer #2: Yes

4. Is the manuscript presented in an intelligible fashion and written in standard English?

Reviewer #1: Yes

Reviewer #2: Yes

5. Review Comments to the Author

Reviewer #1: The authors explore the change in total catch of sea stars from a range of depths at a single site over 22 years. It is an impressive time-series, that explores changes in a group of species over a time period with a known disease epidemic. They find that there were marked declines in total catch of sea stars in 2006 and 2015, with some variability in declines across depths.

My main issue is that the analysis and results for this manuscript focus on ‘total sea star catch’. While I think this can be an interesting question and a different way to look at the trends over time, the multitude of relevant phenotypic differences between the sea stars included in this metric makes it important to be careful in how the results are interpreted. In the species included we have a wide range of average density, rarity, susceptibility to disease, trophic level, competitive ability, etc. In many cases these species are known to interact, and particularly known to interact around responses to sea star wasting disease. I still think that this work is important and should be shared, but I think that there should be some substantial editing to the framing around the work. I have highlighted specific sections below in the line edits where changes could be made to address this issue.

Line edits:

Line 33-37- While these results do indicate that the pattern is for ‘overall abundance of sea stars’, I think you need to be clear in the methods that this measure comes from trends that are measuring the ‘total catch’. It would be possible to make these same statements and have underlying data that was analyzed by species, and that was the assumption that I made on my first read through the abstract. It left me startled when I first looked at your methods and results.

Line 65 – a technical point, but you should use the term ‘signs of disease’ instead of symptoms here. Symptoms are reported, and since invertebrates can’t report, they can’t have symptoms.

Line 98-100- it seems like a citation or two would be appropriate here

Line 117 – between the two sentences on this line, you should clarify that you will be looking at the community as a whole (total catch) and how this will influence the hypotheses you lay out afterwards. Specifically, I think it would be worth mentioning why you would expect the whole community abundance to go down, when not all species were equally affected by SSWD. That could still be a viable hypothesis, but would require the assumption that the dominant species of the communities surveyed were most responsive to SSWD (which certainly is possible).

Line 172 – I am surprised to see that all sea stars were lumped into one category here (total catch), especially since they were identified to species (line 143-144). I appreciate that you likely don’t have enough observations of each species to do this analysis by species, but I think the choice to lump all sea stars together needs to be defended and discussed in the methods. I also think it should be introduced in the methods more explicitly, and the consequences of the lumping explored. How the total catch of sea stars is changing over time is a legitimate and interesting question, but is fundamentally different from how each species responded.

Line 271 – I think rephrasing this to indicate that when you evaluated across all depths you found a changepoint in 2006. At the moment this statement could indicate that you found that change point in each individual depth, which the next sentences clarify that you don’t.

Line 282-287 – there needs to be some discussion here of the fact that you have lumped all sea stars into one category, so the variable responses to wasting might be swamping out the signal. I recognize that you go on in the discussion to discuss the influence of the different species on the response, but I think you need something in this first paragraph to clarify what inference we can take away from these patterns.

Line 302-303 – I think a sentence similar to this should also be in the methods, explaining why you chose to look at catch instead of individual species response.

Line 352-362 – I think it should at least be acknowledged that there is some chance that the decrease in abundance over time could be related to the repeated bottom trawling over years in this location.

Table 2 – I doubt that this strongly influenced your results, but it’s a little odd that you’ve lumped a brittle star (Amphipholis pugetana) in with your total sea star catch. If you are going to keep the brittle star in the data, it should be noted in the methods that you have total asteroid and ophiuroid catch. Assuming of course that you would have identified any brittle star that was caught and that the only one collected was this species.

Reviewer #2: Review of Casendino – PONE-D-22-2571

“Two decades of change in sea star abundance at a subtidal site in Puget Sound, Washington”

General comments: This study benefits from three key aspects. First, as the author’s point out, it leverages a rare and fortunate long-term timeseries. Secondly, the analyses are excellent. Third, it is an impressive use of a dataset generated from a field sampling project for a course. However, the study suffers from writing that doesn’t provide a strong conceptual motivation, and insufficient depth and breadth to provide a convincing story.

Taken on face value (i.e. trends in all species combined), the analyses strongly suggest that factors other than SSWD are responsible for the trajectories of sea stars across depths over time. If the authors quantitatively evaluated these alternative explanations and found stronger relationships, this argument would not only be more compelling, but also be the basis of an interesting story, which is exactly how they close the Discussion section. This could have been the overarching theme of the article, “Contrary to other regions of the West Coast-wide SSWD epidemic, other potential environmental drivers (potential because they are all correlative), overwhelm the relative impact of SSWD in an assemblage of sea stars in Puget Sound.” That is a very interesting results, but requires more analyses.

The other problem I have with the manuscript is the lack of information (temporal trends by depth) for each species. Yes, they are insufficient for rigorous statistical analysis, but not knowing how the different species are potentially contributing to trends is extremely frustrating. The authors are not providing information at the species level that readers would find intriguing.

Overall, because the manuscript describes a very localized phenomenon with little broader conceptual contributions, I think the manuscript is more appropriate for a more regional or disciplinary-focused journal. A marine or environmental journal whose readers would appreciate that wide-scale manifestations of a disease were overwhelmed locally by environmental degradation is something I’d be keen to cite.

Specific comments:

Introduction

I deeply appreciate the emphasis on the value of long-term data sets, but it is not the point of the article. Start the Intro focused on the greater conceptual issue. If the overall conclusion is how local environmental conditions can overwhelm the effects of otherwise disastrous widescale perturbations like epidemics, start with that and how this knowledge advances our understanding of the causes of spatial variation in such perturbations.

Certainly include the point of how long term datasets are central to addressing this issue and also consider additional citations. Especially look for the most recent articles by Lindenmayer:

- Hughes, B.B., Beas-Luna, R., Barner, A.K., Brewitt, K., Brumbaugh, D.R., Cerny-Chipman, E.B., Close, S.L., Coblentz, K.E., De Nesnera, K.L., Drobnitch, S.T. and Figurski, J.D., 2017. Long-term studies contribute disproportionately to ecology and policy. BioScience, 67(3), pp.271-281.

- Lindenmayer DB, Likens GE, Krebs CJ, Hobbs RJ. 2010. Improved probability of detection of ecological “surprises.” Proceedings of the National Academy of Sciences 107: 21957–21962.

- Lindenmayer DB, et al. 2012. Value of long-term ecological studies. Austral Ecology 37: 745–757.

The purpose of the paragraphs describing the significance of the SSWD should be terse and describe the timing, geographic scope and species impacted, pointing out that it occurred in the intertidal and subtidal. Point made and no greater detail is necessary. There are several articles that describe this phenomenon for subtidal (especially, P. helanthoides) ecosystems, even though they also draw attention to the role of the MHW. They should be cited as they contribute to geographic scope and ecological consequences associated, in part, to SSWD.

- McPherson, M.L., Finger, D.J., Houskeeper, H.F., Bell, T.W., Carr, M.H., Rogers-Bennett, L. and Kudela, R.M., 2021. Large-scale shift in the structure of a kelp forest ecosystem co-occurs with an epizootic and marine heatwave. Communications biology, 4(1), pp.1-9.

- Smith, J.G., Tomoleoni, J., Staedler, M., Lyon, S., Fujii, J. and Tinker, M.T., 2021. Behavioral responses across a mosaic of ecosystem states restructure a sea otter–urchin trophic cascade. Proceedings of the National Academy of Sciences, 118(11), p.e2012493118.

-

Line 77- “Eisenlord et al. (2016) tracked 78 SSWD prevalence in P. ochraceus in Washington State from December 2013 to July 2015 [17].” Summarize this entire study in 1-2 sentences as one of several examples of spatial variation in the population consequence of SSWD, which is the key point of this article (i.e. that it was not manifest strongly at the study site).

Methods

Line 139; “Each year, five trawls took place at each depth, 140 corresponding to discrete time periods: early morning (~6:00–8:00), morning (~10:00–12:00), 141 afternoon (~15:00–17:00), evening (~20:00–22:00), and night (~1:00–3:00). Each trawl sampled 142 benthic habitat for ~5 min over 370 m (range = 367 to 538 m).”

Please make clearer the spatial distribution of transects. Was essentially the same transect area sampled repeatedly over time on a sampling day, or was the location of the transects at a given depth stratum offset so as not to sample the very same area over time? We’re you depleting the stars over the sequence of the samples and summing them across samples? Did catch at each depth decline over the day?

Results

Table 2: “listed in order of overall abundance”. To make this more obvious and informative, add a column for mean abundance across all depth strata for each species.

Line 299: “The presence of relatively resilient D. imbricata in shallow subtidal depths alone may also have offset localized mortality rates, perhaps explaining why high abundances of P. helianthoides (and SSWD-associated mortality) in the shallow subtidal were not associated with more drastic declines in those depths. Unfortunately, replication at the level of species was insufficient to test for species-specific trends in our dataset.”

Yes, you can’t statistically evaluate change for each species by depth, but not seeing graphically how each species changed at each depth doesn’t allow the reader (or you) to evaluate the potential contributions of each species to the trends of combined species and that is frustrating. You need to make supplemental graphs of the trends in density for each species by each depth stratum. As the authors indicate, P. helianthoides might have tanked at shallower depths but it is masked by lack of change in D. imbricata. Readers need to see the trends for each species, as poor as they are because of such low densities, to interpret the trends in the combined species.

Line 325: “In the context of these reports, our 326 results strongly suggest a lagged onset of SSWD-associated mortality at Port Madison.” But again, it is frustrating not to know which species contributed most to this pattern. What if the major contributors were species thought not to be susceptible to SSWD?

Line 335: “… suggesting that declines may have resulted from a deleterious environmental variable. Since 1999, the frequency of phytoplankton blooms and average chlorophyll-a in Puget Sound have declined;…”

An analysis of how well these other environmental variables explained the observed population trend would be very interesting. The take home of such a study would be “Yeah, SSWD was devastating in many parts of the West Coast, but environmental degradation (or natural long-term trends) in Puget Sound overwhelmed such effects!”. That’s interesting!

Line 343: “we expected high temperatures to be correlated with low sea star density…” Yes, but your points as to why higher temps could be detrimental (e.g.,. hypoxia) beg the question of whether any species were migrating across depth zones either to avoid detrimental conditions or in pursuit of more beneficial conditions (prey availability). How species number changed across depth zones over time would be interesting and might explain these surprising relationships.

Line 367: This work suggests that subtidal sea star communities experienced gradual and steep

declines not strictly attributable to the SSWD epizootic of 2013, warranting further examination

of other external threats posed to such communities in the Salish Sea.” That says it all, and should be the title and point of this article.

6. PLOS authors have the option to publish the peer review history of their article (what does this mean?). If published, this will include your full peer review and any attached files.

Reviewer #1: No

Reviewer #2: No

---

## [Author Response · Author response to Decision Letter 0]

2 May 2023

Academic Editor Comments to Author

Thank you for submitting your manuscript to PLOS ONE. After careful consideration, we feel that it has merit but does not fully meet PLOS ONE’s publication criteria as it currently stands. Therefore, we invite you to submit a revised version of the manuscript that addresses the points raised during the review process.

The two reviewers have made many good suggestions for improving your manuscript. I'll point out that both of them were not satisfied with lumping everything into total sea stars, and wanted species specific trends wherever possible. Given the low densities (Table 2), the absolute numbers must be low also. A figure with the species-specific numbers could be a good idea. The Github link to the data that you provided isn't complete, so I couldn't look at the direct numbers. If I am doing the math correctly converting from Table 2, lots of those values are 1, 2, etc. I wonder if you could do some kind of heat map with the absolute values? That might be an easier way to display the species-specific trends in an informative manner. 

Many thanks for facilitating this thorough, thoughtful round of peer review. The reviewers have made a number of very helpful suggestions for improving the manuscript. We wholeheartedly agree that transparent reporting on species-specific trends is necessary to contextualize our results regarding trends in total catch of sea stars over time. You make an excellent point about bringing in a visualization of absolute abundance to illustrate species-specific trends overtime. To this end, we have added in Figure 2 (see below), which depicts total sea star catch in a stacked bar plot. Species composition at any given depth and year is clearly represented, illuminating trends in the abundances of individual species. 

As we discuss in further detail below, we have also modified our analysis to account for differences in susceptibility to SSWD among species. We are extra grateful to the reviewers for this suggestion, because the results clearly show that high-susceptibility sea stars did in fact respond to the 2013-2015 SSWD event, whereas moderate-susceptibility sea stars did not. We believe that this approach goes far in addressing the reviewers’ concerns about capturing the contributions of certain species to observed trends in total sea star abundance, particularly in the context of variable responses to SSWD.

Fig 2. Observed sea star catch by species at four depths from 1997 through 2019. 

Sea star catch per year, summarized per species over 5 tows within a 24-h period each year at each depth from 1997-2019. On average, trawl tow distance was 372 m, with values ranging from 367 to 538 m.

Please ensure that your manuscript meets PLOS ONE's style requirements, including those for file naming. The PLOS ONE style templates can be found at https://journals.plos.org/plosone/s/file?id=wjVg/PLOSOne_formatting_sample_main_body.pdf and https://journals.plos.org/plosone/s/file?id=ba62/PLOSOne_formatting_sample_title_authors_affiliations.pdf.

Thank you for providing these resources. The content and naming of submitted files should now be in accordance with PLOS ONE’s style requirements. 

In your Methods section, please provide additional information regarding the permits you obtained for the work. Please ensure you have included the full name of the authority that approved the field site access and, if no permits were required, a brief statement explaining why.

Line 134: “The cruise was designed as a teaching experience for students enrolled in a University of Washington course (Fisheries Ecology / FISH 312) [43]. The sampling protocol for these field trips received a full review every three years from the School of Aquatic and Fishery Sciences and the Institutional Animal Care and Use Committee of the University of Washington (IACUC protocol # 2442-13). In addition, the field trips were annually reviewed and permitted by the Washington Department of Fish and Wildlife (SCP 22-102).”

We note that you have stated that you will provide repository information for your data at acceptance. Should your manuscript be accepted for publication, we will hold it until you provide the relevant accession numbers or DOIs necessary to access your data. If you wish to make changes to your Data Availability statement, please describe these changes in your cover letter and we will update your Data Availability statement to reflect the information you provide.

We will provide repository links upon acceptance and make sure to edit the Data Availability statement appropriately.

We note that [Figure 1] in your submission contain [map/satellite] images which may be copyrighted. All PLOS content is published under the Creative Commons Attribution License (CC BY 4.0), which means that the manuscript, images, and Supporting Information files will be freely available online, and any third party is permitted to access, download, copy, distribute, and use these materials in any way, even commercially, with proper attribution. For these reasons, we cannot publish previously copyrighted maps or satellite images created using proprietary data, such as Google software (Google Maps, Street View, and Earth). For more information, see our copyright guidelines: http://journals.plos.org/plosone/s/licenses-and-copyright.

Thank you for pointing this out. We have removed all copyrighted imagery from Figure 1. This figure now includes public domain imagery from the National Agriculture Imagery Program (USDA) and the Landsat Program (USGS). 

Reviewer #1:

Is the manuscript technically sound, and do the data support the conclusions?: Partly

Has the statistical analysis been performed appropriately and rigorously?: Yes

Have the authors made all data underlying the findings in their manuscript fully available?: Yes

Is the manuscript presented in an intelligible fashion and written in standard English?: Yes

General Comments: 

The authors explore the change in total catch of sea stars from a range of depths at a single site over 22 years. It is an impressive time-series that explores changes in a group of species over a time period with a known disease epidemic. They find that there were marked declines in total catch of sea stars in 2006 and 2015, with some variability in declines across depths.

My main issue is that the analysis and results for this manuscript focus on ‘total sea star catch’. While I think this can be an interesting question and a different way to look at the trends over time, the multitude of relevant phenotypic differences between the sea stars included in this metric makes it important to be careful in how the results are interpreted. In the species included we have a wide range of average density, rarity, susceptibility to disease, trophic level, competitive ability, etc. In many cases these species are known to interact, and particularly known to interact around responses to sea star wasting disease. I still think that this work is important and should be shared, but I think that there should be some substantial editing to the framing around the work. I have highlighted specific sections below in the line edits where changes could be made to address this issue.

Many thanks for your thoughtful suggestions. We agree that our interpretation of results regarding total catch of sea stars over time should have included further discussion of how individual species may have contributed to the trends we observed, given the myriad ways in which represented species differ (as you mention). To tease out the influence of variability among species, and more specifically, variation in susceptibility to wasting, on temporal trends in sea star abundance, we analyzed the total catch of species most susceptible to wasting separately from the total catch of less susceptible species. We are extra grateful to the reviewers for this suggestion, because the results clearly show that high-susceptibility sea stars did in fact respond to the 2013-2015 SSWD event, whereas moderate-susceptibility sea stars did not. We believe that these new findings highlight ways in which the effects of ecological catastrophes (e.g., SSWD) or long-term cycles/disturbances are mediated by differences among species. 

Specific Comments: 

Line 33-37- While these results do indicate that the pattern is for ‘overall abundance of sea stars’, I think you need to be clear in the methods that this measure comes from trends that are measuring the ‘total catch’. It would be possible to make these same statements and have underlying data that was analyzed by species, and that was the assumption that I made on my first read through the abstract. It left me startled when I first looked at your methods and results.

Thank you, we agree that this needed further clarification in the abstract. We have modified the abstract text to accurately reflect our approach.

Line 24: “We used two decades (1997–2019) of scientific trawling data from a subtidal, benthic site in Puget Sound, Washington, USA to test for gradual trends and sudden shifts in total sea star abundance across 11 species.”

Line 28: “To account for species-level differences in SSWD susceptibility, we divided our sea star abundance data into two categories, depending on the extent to which the species is susceptible to SSWD, then conducted parallel analyses for high-susceptibility and moderate-susceptibility species. The abundance of high-susceptibility sea stars declined in 2014 across depths. In contrast, the abundance of moderate-susceptibility species trended downward throughout the years at the deepest depths – 50 and 70 m – and suddenly declined in 2006 across depths. Water temperature was positively correlated with the abundance of moderate-susceptibility species, and uncorrelated with high-susceptibility sea star abundance.”

We have also modified the language in the Methods section to reflect that our measure of high-susceptibility and moderate-susceptibility species density is based on total catch across these species groupings.

Line 70: “We then divided the total number of sea stars (within each susceptibility category) found in each trawl tow (corresponding to a single depth and time) by the area swept (and multiplied by 1000 for readability) to calculate total sea star density (specimens per 1000 m2) of high-susceptibility and moderate-susceptibility species.”

Line 65 – a technical point, but you should use the term ‘signs of disease’ instead of symptoms here. Symptoms are reported, and since invertebrates can’t report, they can’t have symptoms.

Thank you for catching this mistake. Throughout the manuscript, we have replaced occurrences of the word “symptom” with “signs of disease.”

Line 98-100- it seems like a citation or two would be appropriate here

This sentence has been removed from the manuscript.

Line 117 – between the two sentences on this line, you should clarify that you will be looking at the community as a whole (total catch) and how this will influence the hypotheses you lay out afterwards. Specifically, I think it would be worth mentioning why you would expect the whole community abundance to go down, when not all species were equally affected by SSWD. That could still be a viable hypothesis, but would require the assumption that the dominant species of the communities surveyed were most responsive to SSWD (which certainly is possible).

Thank you for this insightful comment. It was inaccurate to assume that all species represented in our data would have been impacted similarly by the onset of SSWD. In this section, we have now added text explaining that we expect SSWD’s effects on trends of sea star abundance to differ based on the susceptibility of represented species to SSWD.

Line 116: “We predicted declines in sea star density coincident with the 2013–2015 SSWD event, with a possible recovery from 2016 onwards [35]. We further predicted that these declines would be steeper for species with high reported susceptibility to SSWD relative to species with moderate reported susceptibility, and that warm temperatures would be associated with declines, as might result from increased microbial activity or eroded sea star immune defenses [14,36,39].”

Line 172 – I am surprised to see that all sea stars were lumped into one category here (total catch), especially since they were identified to species (line 143-144). I appreciate that you likely don’t have enough observations of each species to do this analysis by species, but I think the choice to lump all sea stars together needs to be defended and discussed in the methods. I also think it should be introduced in the methods more explicitly, and the consequences of the lumping explored. How the total catch of sea stars is changing over time is a legitimate and interesting question, but is fundamentally different from how each species responded.

We believe that our modified approach to run analyses in parallel on total catch of high-susceptibility species and total catch of moderate-susceptibility species, and a substantial amount of re-framing in the Introduction and Discussion around differential responses of species to SSWD, will address the issue you have raised here about aggregating species across a gradient of responses to SSWD. See paragraph beginning at Line 157 (Methods section).

Line 157: “We categorized our data by the relative susceptibility of sea star species to SSWD based on Schiebelut et al. (2023), who classified the level of SSWD-associated mortality faced by several sea star species [21]. Of the species present in our data, five were categorized by Schiebelut et al. as “high mortality” (Solaster stimpsoni, S. dawsoni, Pycnopodia helianthoides, Pisaster brevispinus, E. troschelii), which we categorized as “high-susceptibility” species. Two additional species were categorized by Schiebelut et al. as “noticeable mortality” (D. imbricata, Henricia leviuscula) and four as “likely affected” (Mediaster aequalis, Luidia foliolata, Crossaster papposus, Hippasteria spinosa), all of which we categorized as “moderate-susceptibility” species, which we assumed would experience lower SSWD-associated mortality than high-susceptibility species. Subsequent processing and analysis of abundance data was conducted in parallel for these two data groupings. We did not analyze species-specific trends in our data due to insufficient replication at the level of individual species.”

Line 271 – I think rephrasing this to indicate that when you evaluated across all depths you found a changepoint in 2006. At the moment this statement could indicate that you found that change point in each individual depth, which the next sentences clarify that you don’t.

Thank you for your attention to detail on this point. We have reworded this part of the Results section to more clearly reflect changepoints identified at depth versus those identified when evaluating our data across depth. 

Line 311: “Within and across all depth categories, we observed no significant changepoints in high-susceptibility sea star catch (Fig 5). We observed a significant changepoint in moderate-susceptibility sea star catch in 2006 when we evaluated our data across all depths (Fig 6). When depths were analyzed individually for this subset of species, a changepoint was observed in 2006 at 70 m.”

Line 282-287 – there needs to be some discussion here of the fact that you have lumped all sea stars into one category, so the variable responses to wasting might be swamping out the signal. I recognize that you go on in the discussion to discuss the influence of the different species on the response, but I think you need something in this first paragraph to clarify what inference we can take away from these patterns.

We have modified the first paragraph of the Discussion to accommodate our new approach, which we believe removes the problem associated with lumping all sea stars into one category and allows us to more confidently interpret observed trends. Our findings regarding trends of high-susceptibility sea star species abundance now agree with previous work pointing to marked declines of particular species following SSWD’s onset. 

Line 327: “Consistent with our predictions and previous reports in Washington State [14,32,53], our analyses suggest that the density of high-susceptibility sea stars was lower after 2013 (Fig 3). Unexpectedly, our analyses also suggest that moderate-susceptibility sea stars experienced long-term declines throughout the study period in deeper water (50 and 70 m) (Fig 4), and a sudden decline in 2006 at 70 m and across depths (Fig 6).”

Line 302-303 – I think a sentence similar to this should also be in the methods, explaining why you chose to look at catch instead of individual species response.

Thank you for this suggestion. We have included the following sentence in the Methods section.

Line 167: “We did not analyze species-specific trends in our data due to insufficient replication at the level of individual species.”

Line 352-362 – I think it should at least be acknowledged that there is some chance that the decrease in abundance over time could be related to the repeated bottom trawling over years in this location.

We agree that this should be mentioned as a potential explanation for observed declines in sea star catch over time. However, we believe this to be an unlikely explanation, given that we found no evidence of serial depletion over consecutive trawls at the same depth in a given sampling year. We have now addressed this issue in the Methods and Discussion sections.

Line 199: “While trawls were conducted in roughly the same location for each depth sampled, natural variability in each trawl’s path made overlap unlikely. We confirmed this using a generalized linear model where expected catch was modeled as a function of trawl number (corresponding to time of day) at each depth. We observed no significant relationship between trawl number and expected total sea star catch (estimate = 0.03, ± SE = 0.12, p = 0.79).”

Line 367: “It is possible that the observed long-term declines in sea star abundance resulted from our repeated bottom trawling at a study site over multiple years but this is unlikely. Natural variability in each trawl’s path would make it unlikely for the same area to be trawled twice, and we found no evidence of serial depletion across consecutive trawls within a given sampling year.”

Table 2 – I doubt that this strongly influenced your results, but it’s a little odd that you’ve lumped a brittle star (Amphipholis pugetana) in with your total sea star catch. If you are going to keep the brittle star in the data, it should be noted in the methods that you have total asteroid and ophiuroid catch. Assuming of course that you would have identified any brittle star that was caught and that the only one collected was this species.

Thank you for pointing out this error! We intended to include only sea stars in our analyses. We have now omitted Amphipholis pugetana from our sample. 

Reviewer #2:

Is the manuscript technically sound, and do the data support the conclusions?: Partly

Has the statistical analysis been performed appropriately and rigorously?: Yes

Have the authors made all data underlying the findings in their manuscript fully available?: Yes

Is the manuscript presented in an intelligible fashion and written in standard English?: Yes

General Comments: 

This study benefits from three key aspects. First, as the author’s point out, it leverages a rare and fortunate long-term time series. Secondly, the analyses are excellent. Third, it is an impressive use of a dataset generated from a field sampling project for a course. However, the study suffers from writing that doesn’t provide a strong conceptual motivation, and insufficient depth and breadth to provide a convincing story.

Taken on face value (i.e. trends in all species combined), the analyses strongly suggest that factors other than SSWD are responsible for the trajectories of sea stars across depths over time. If the authors quantitatively evaluated these alternative explanations and found stronger relationships, this argument would not only be more compelling, but also be the basis of an interesting story, which is exactly how they close the Discussion section. This could have been the overarching theme of the article, “Contrary to other regions of the West Coast-wide SSWD epidemic, other potential environmental drivers (potential because they are all correlative), overwhelm the relative impact of SSWD in an assemblage of sea stars in Puget Sound.” That is a very interesting results, but requires more analyses.

The other problem I have with the manuscript is the lack of information (temporal trends by depth) for each species. Yes, they are insufficient for rigorous statistical analysis, but not knowing how the different species are potentially contributing to trends is extremely frustrating. The authors are not providing information at the species level that readers would find intriguing.

Overall, because the manuscript describes a very localized phenomenon with little broader conceptual contributions, I think the manuscript is more appropriate for a more regional or disciplinary-focused journal. A marine or environmental journal whose readers would appreciate that wide-scale manifestations of a disease were overwhelmed locally by environmental degradation is something I’d be keen to cite.

Thank you for these insightful comments and thoughtful revision of our manuscript. We totally agree that an investigation into environmental drivers of sea star abundance trends would be valuable and interesting to readers. However, our new analyses (conducted at your suggestion) show a pattern consistent with previously reported sea star declines associated with SSWD. Therefore, we believe that the existing framing now fits well with the results.

To tease out the influence of variability among species, and more specifically, variation in susceptibility to wasting, on temporal trends in sea star abundance, we analyzed the total catch of species most susceptible to wasting separately from the total catch of less susceptible species. Our new results clearly show that high-susceptibility sea stars did in fact respond to the 2013-2015 SSWD event, whereas moderate-susceptibility sea stars did not. 

With these new findings, we believe that the study’s original motivation – to capture temporal variation in Port Madison’s subtidal sea star community over two decades of change, overlapping with a marine epizootic affecting sea stars, by leveraging valuable long-term data – provides a strong framework for informing our understanding of subtidal sea star community dynamics. While declines in the abundance of moderate-susceptibility species predating SSWD’s onset remain unexplained, we believe that such findings have stand-alone merit as documented evidence of natural variability or a protracted anthropogenic disturbance. 

We have also worked to address your feedback regarding the need for more transparent reporting of temporal trends by depth for each species by adding Figure 2 (see below), which depicts total sea star catch in a stacked bar plot. Species composition at any given depth and year is clearly represented, illuminating species-specific abundance trends. 

Fig 2. Observed sea star catch by species at four depths from 1997 through 2019. 

Sea star catch per year, summarized per species over 5 tows within a 24-h period each year at each depth from 1997-2019. On average, trawl tow distance was 372 m, with values ranging from 367 to 538 m.

Specific Comments: 

Introduction

I deeply appreciate the emphasis on the value of long-term data sets, but it is not the point of the article. Start the Intro focused on the greater conceptual issue. If the overall conclusion is how local environmental conditions can overwhelm the effects of otherwise disastrous wide scale perturbations like epidemics, start with that and how this knowledge advances our understanding of the causes of spatial variation in such perturbations.

Certainly include the point of how long term datasets are central to addressing this issue and also consider additional citations. Especially look for the most recent articles by Lindenmayer:

- Hughes, B.B., Beas-Luna, R., Barner, A.K., Brewitt, K., Brumbaugh, D.R., Cerny-Chipman, E.B., Close, S.L., Coblentz, K.E., De Nesnera, K.L., Drobnitch, S.T. and Figurski, J.D., 2017. Long-term studies contribute disproportionately to ecology and policy. BioScience, 67(3), pp.271-281.

- Lindenmayer DB, Likens GE, Krebs CJ, Hobbs RJ. 2010. Improved probability of detection of ecological “surprises.” Proceedings of the National Academy of Sciences 107: 21957–21962.

- Lindenmayer DB, et al. 2012. Value of long-term ecological studies. Austral Ecology 37: 745–757.

We have incorporated the useful references you mention into the Introduction. See line 48.

Line 48: “Long-term datasets are critical for environmental policy, management, and conservation [2,3], providing insight by revealing shifts from one stable pattern to another [4] and revealing the effects of long-term environmental trends such as climate change [5]. Furthermore, long-term datasets generate baseline data that can document historical ecosystem states in the event of sudden ecological changes or catastrophes (e.g., epizootics) [6,7].”

We certainly agree that an investigation into environmental drivers of sea star abundance trends would be valuable and interesting to readers. However, we believe that the existing framing now fits well with our results showing a pattern consistent with previously reported sea star declines associated with SSWD. 

With these new findings, we believe that the study’s original motivation – to capture temporal variation in Port Madison’s subtidal sea star community over two decades of change, overlapping with a marine epizootic affecting sea stars, by leveraging valuable long-term data – provides a strong framework for informing our understanding of subtidal sea star community dynamics. While declines in the abundance of moderate-susceptibility species predating SSWD’s onset remain unexplained, we believe that such findings have stand-alone merit as documented evidence of natural variability or a protracted anthropogenic disturbance. 

The purpose of the paragraphs describing the significance of the SSWD should be terse and describe the timing, geographic scope and species impacted, pointing out that it occurred in the intertidal and subtidal. Point made and no greater detail is necessary. There are several articles that describe this phenomenon for subtidal (especially, P. helanthoides) ecosystems, even though they also draw attention to the role of the MHW. They should be cited as they contribute to geographic scope and ecological consequences associated, in part, to SSWD.

- McPherson, M.L., Finger, D.J., Houskeeper, H.F., Bell, T.W., Carr, M.H., Rogers-Bennett, L. and Kudela, R.M., 2021. Large-scale shift in the structure of a kelp forest ecosystem co-occurs with an epizootic and marine heatwave. Communications biology, 4(1), pp.1-9.

- Smith, J.G., Tomoleoni, J., Staedler, M., Lyon, S., Fujii, J. and Tinker, M.T., 2021. Behavioral responses across a mosaic of ecosystem states restructure a sea otter–urchin trophic cascade. Proceedings of the National Academy of Sciences, 118(11), p.e2012493118.

We’ve now reworked the Introduction to introduce SSWD in a concise manner while still providing key details of its impact on sea star communities and ecosystem structure. We have included these references in the Introduction and thank the reviewer for pointing us to such relevant resources. See paragraphs beginning at Line 61 (description of SSWD) and Line 72 (community and ecosystem impacts of SSWD). 

Line 61: “Sea-star wasting disease (SSWD) – an epizootic that began in 2013 and affected dozens of species along ~5,000 km of North America’s west coast [13,14] – covered a larger area and resulted in greater mass mortality than other sea star wasting events in recent history [15,16], and coincided with a marine heatwave that lasted from 2013 to 2016 [17,18]. The sunflower star (Pycnopodia helianthoides) experienced substantial SSWD-associated declines [13,14], resulting in its designation as “Critically Endangered” by the International Union for Conservation of Nature in 2020 [19]. Miner et al. (2018) observed precipitous declines in ochre star (Pisaster ochraceus) abundance from southern California to southeastern Alaska in 2014 and 2015 [20]; in Oregon, Menge et al. (2016) found that P. ochraceus biomass declined 80–99% during the same period [15], though P. ochraceus recruitment has increased in the years following SSWD’s onset [21].”

Line 72: “SSWD altered intertidal and subtidal community structure across the west coast of North America. Pycnopodia helianthoides and Pisaster ochraceus are keystone predators that exert strong top-down control of other species in subtidal and intertidal habitats [22-26]. Schultz et al. (2016) found that the SSWD-induced decline of predatory P. helianthoides was followed by a four-fold increase in green sea urchin (Strongylocentrotus droebachiensis) abundance in the Salish Sea, and a decline in kelp cover from 4% (± 10%) of the study area to < 1% (± 2%) [24]. Changes to kelp forests in California during this time period also indicated the influence of P. helianthoides declines on urchin-mediated regime shifts [17,27]. Shifts in ecosystem dynamics attributable to SSWD also include changes to sea star community structure. Montecino-Latorre et al. (2016) monitored sea star density in depths of 6–18 m [28] in scuba diving surveys and observed that declines in pink sea stars (Pisaster brevispinus) and Pycnopodia helianthoides were coupled with an increase in leather stars (Dermasterias imbricata) but no change in vermillion stars (Mediaster spp.) following the 2013 outbreak. Additionally, Kay et al. (2019) observed that mottled stars (Evasterias troschelii) became dominant following SSWD’s onset as formerly dominant Pisaster ochraceus declined, potentially due to competitive release [29].”

Line 77- “Eisenlord et al. (2016) tracked 78 SSWD prevalence in P. ochraceus in Washington State from December 2013 to July 2015 [17].” Summarize this entire study in 1-2 sentences as one of several examples of spatial variation in the population consequence of SSWD, which is the key point of this article (i.e. that it was not manifest strongly at the study site).

We agree with the reviewer that we should have highlighted spatial variation in SSWD impact as a notable potential explanation of why total sea star abundance did not suddenly decline in 2014 at our study site. However, our new analyses of the total catch of high-susceptibility species suggest that certain species did face declines attributable to the SSWD epizootic in Puget Sound, echoing the findings of Eisenlord et al. (2016), which we reference in the text at Line 337. Given our study’s small spatial scale and results consistent with reports of SSWD in the study area, we no longer see spatial variability of SSWD impact as a key part of the manuscript. 

Methods

Line 139; “Each year, five trawls took place at each depth, 140 corresponding to discrete time periods: early morning (~6:00–8:00), morning (~10:00–12:00), 141 afternoon (~15:00–17:00), evening (~20:00–22:00), and night (~1:00–3:00). Each trawl sampled 142 benthic habitat for ~5 min over 370 m (range = 367 to 538 m).”

Please make clearer the spatial distribution of transects. Was essentially the same transect area sampled repeatedly over time on a sampling day, or was the location of the transects at a given depth stratum offset so as not to sample the very same area over time? We’re you depleting the stars over the sequence of the samples and summing them across samples? Did catch at each depth decline over the day?

Thank you for highlighting the need for more clarity in regarding our sampling approach. We have modified language in the Methods section to clarify our trawling approach, specifying that five trawls took place throughout the day at each depth and occurred in consistent locations each year. Furthermore, we now highlight twice in the manuscript that we found no evidence of serial depletion occurring throughout the day at depth. 

Line 140: “The vessel, which ran each year for two days in mid-May (between 10 and 18 May), towed the SCCWRP net along the bottom in set trawling locations corresponding to four depths: 10, 25, 50 and 70 m (Fig 1). Each year, five trawls took place at each depth, corresponding to discrete time periods: early morning (~6:00–8:00), morning (~10:00–12:00), afternoon (~15:00–17:00), evening (~20:00–22:00), and night (~1:00–3:00).”

Line 146: “Following each trawl, all sea stars caught were identified to species, counted, recorded, and released.”

Line 199: “While trawls were conducted in roughly the same location for each depth sampled, natural variability in each trawl’s path made overlap unlikely. We confirmed this using a generalized linear model where expected catch was modeled as a function of trawl number (corresponding to time of day) at each depth. We observed no significant relationship between trawl number and expected total sea star catch (estimate = 0.03, ± SE = 0.12, p = 0.79).”

Line 369: “Natural variability in each trawl’s path would make it unlikely for the same area to be trawled twice, and we found no evidence of serial depletion across consecutive trawls within a given sampling year.”

Results

Table 2: “listed in order of overall abundance”. To make this more obvious and informative, add a column for mean abundance across all depth strata for each species.

We have modified Table 2 to include a column, titled “All depths,” with mean abundance across all depths for each species. 

Line 299: “The presence of relatively resilient D. imbricata in shallow subtidal depths alone may also have offset localized mortality rates, perhaps explaining why high abundances of P. helianthoides (and SSWD-associated mortality) in the shallow subtidal were not associated with more drastic declines in those depths. Unfortunately, replication at the level of species was insufficient to test for species-specific trends in our dataset.”

Yes, you can’t statistically evaluate change for each species by depth, but not seeing graphically how each species changed at each depth doesn’t allow the reader (or you) to evaluate the potential contributions of each species to the trends of combined species and that is frustrating. You need to make supplemental graphs of the trends in density for each species by each depth stratum. As the authors indicate, P. helianthoides might have tanked at shallower depths but it is masked by lack of change in D. imbricata. Readers need to see the trends for each species, as poor as they are because of such low densities, to interpret the trends in the combined species.

We completely agree that species-specific trends should be clearly represented to contextualize trends of total sea star abundance. We have added in Figure 2 (see above), which depicts total sea star catch in a stacked bar plot. Species composition at any given depth and year is clearly represented, illuminating trends in the abundances of individual species. 

Line 325: “In the context of these reports, our results strongly suggest a lagged onset of SSWD-associated mortality at Port Madison.” But again, it is frustrating not to know which species contributed most to this pattern. What if the major contributors were species thought not to be susceptible to SSWD?

We have modified our analysis approach to analyze high-susceptibility and moderate-susceptibility species separately and added a new figure (Figure 2) depicting catch of individual species over time at depth. We believe that these actions address the heart of the issue raised here by highlighting the influence of variability among species on observed trends. 

Line 335: “… suggesting that declines may have resulted from a deleterious environmental variable. Since 1999, the frequency of phytoplankton blooms and average chlorophyll-a in Puget Sound have declined;…”

An analysis of how well these other environmental variables explained the observed population trend would be very interesting. The take home of such a study would be “Yeah, SSWD was devastating in many parts of the West Coast, but environmental degradation (or natural long-term trends) in Puget Sound overwhelmed such effects!”. That’s interesting!

We certainly agree that an investigation into environmental drivers of sea star abundance trends would be valuable and interesting to readers. However, we believe that the existing framing now fits well with our results showing a pattern consistent with previously reported sea star declines associated with SSWD. 

With these new findings, we believe that the study’s original motivation – to capture temporal variation in Port Madison’s subtidal sea star community over two decades of change, overlapping with a marine epizootic affecting sea stars, by leveraging valuable long-term data – provides a strong framework for informing our understanding of subtidal sea star community dynamics. While declines in the abundance of moderate-susceptibility species predating SSWD’s onset remain unexplained, we believe that such findings have stand-alone merit as documented evidence of natural variability or a protracted anthropogenic disturbance. 

Line 378: “While outside of the scope of this study, complex interactions of environmental variables and their effects on wasting prevalence in sea stars—isolated from the influence of any pathogens linked to SSWD—warrant further investigation. For species less susceptible to SSWD, the long-term effects of environmental and/or anthropogenic factors seemed to have overwhelmed the effects of the 2013 epizootic.”

Line 343: “we expected high temperatures to be correlated with low sea star density…” Yes, but your points as to why higher temps could be detrimental (e.g.,. hypoxia) beg the question of whether any species were migrating across depth zones either to avoid detrimental conditions or in pursuit of more beneficial conditions (prey availability). How species number changed across depth zones over time would be interesting and might explain these surprising relationships.

We thank the reviewer for this insightful comment. However, upon visualizing the distribution of species at depth over time (see Figure 2), species richness appears to have remained largely consistent across the study period, providing little evidence of species migration across depths. We also may be unable to detect substantial changes in species richness due to small sample sizes throughout the study period. 

Line 367: This work suggests that subtidal sea star communities experienced gradual and steep

declines not strictly attributable to the SSWD epizootic of 2013, warranting further examination

of other external threats posed to such communities in the Salish Sea.” That says it all, and should be the title and point of this article. 

We fully appreciate that an investigation into environmental drivers of sea star abundance trends would be valuable and interesting to readers. However, our new analyses show a pattern consistent with previously reported sea star declines associated with SSWD. Therefore, we believe that the existing framing now fits well with the results.

We believe that the study’s original motivation – to capture temporal variation in Port Madison’s subtidal sea star community over two decades of change, overlapping with a marine epizootic affecting sea stars, by leveraging valuable long-term data – provides a strong framework for informing our understanding of subtidal sea star community dynamics. While declines in the abundance of moderate-susceptibility species predating SSWD’s onset remain unexplained, we believe that such findings have stand-alone merit as documented evidence of natural variability or a protracted anthropogenic disturbance.

---

## [Editor Report · Decision Letter 1]

16 May 2023

Two decades of change in sea star abundance at a subtidal site in Puget Sound, Washington

PONE-D-22-25271R1

Dear Dr. Casendino,

We’re pleased to inform you that your manuscript has been judged scientifically suitable for publication and will be formally accepted for publication once it meets all outstanding technical requirements.

Kind regards,

Erik V. Thuesen, Ph.D.

Academic Editor

PLOS ONE
---

## [Editor Report · Acceptance letter]

2 Jun 2023

PONE-D-22-25271R1 

Two decades of change in sea star abundance at a subtidal site in Puget Sound, Washington 

Dear Dr. Casendino:

I'm pleased to inform you that your manuscript has been deemed suitable for publication in PLOS ONE. Congratulations! Your manuscript is now with our production department. 

Kind regards, 

on behalf of

Dr. Erik V. Thuesen 

Academic Editor

PLOS ONE